# IGF-1 and IGF-2 as Molecules Linked to Causes and Consequences of Obesity from Fetal Life to Adulthood: A Systematic Review

**DOI:** 10.3390/ijms25073966

**Published:** 2024-04-02

**Authors:** Justyna Szydlowska-Gladysz, Adrianna Edyta Gorecka, Julia Stepien, Izabela Rysz, Iwona Ben-Skowronek

**Affiliations:** Department of Pediatric Endocrinology and Diabetology with Endocrine-Metabolic Laboratory, Medical University in Lublin, 20-093 Lublin, Poland

**Keywords:** obesity, IGFs, pediatric obesity, IGF-1, IGF-2, puberty onset, metabolic syndrome, diabetes, metabolic-associated fatty liver disease (MAFLD), cancer development

## Abstract

This study examines the impact of insulin-like growth factor 1 (IGF-1) and insulin-like growth factor 2 (IGF-2) on various aspects of children’s health—from the realms of growth and puberty to the nuanced characteristics of metabolic syndrome, diabetes, liver pathology, carcinogenic potential, and cardiovascular disorders. A comprehensive literature review was conducted using PubMed, with a Preferred Reporting Items for Systematic Reviews and Meta-Analyses (PRISMA) method employing specific keywords related to child health, obesity, and insulin-like growth factors. This study reveals associations between insulin-like growth factor 1 and birth weight, early growth, and adiposity. Moreover, insulin-like growth factors play a pivotal role in regulating bone development and height during childhood, with potential implications for puberty onset. This research uncovers insulin-like growth factor 1 and insulin-like growth factor 2 as potential biomarkers and therapeutic targets for metabolic dysfunction-associated liver disease and hepatocellular carcinoma, and it also highlights the association between insulin-like growth factors (IGFs) and cancer. Additionally, this research explores the impact of insulin-like growth factors on cardiovascular health, noting their role in cardiomyocyte hypertrophy. Insulin-like growth factors play vital roles in human physiology, influencing growth and development from fetal stages to adulthood. The impact of maternal obesity on children’s IGF levels is complex, influencing growth and carrying potential metabolic consequences. Imbalances in IGF levels are linked to a range of health conditions (e.g., insulin resistance, glucose intolerance, metabolic syndrome, and diabetes), prompting researchers to seek novel therapies and preventive strategies, offering challenges and opportunities in healthcare.

## 1. Introduction

Obesity in children is a socially and clinically significant problem. In developed countries, it has reached high levels and consistently been maintained at these over the years. In the pediatric population, obesity is diagnosed based on the Body Mass Index (BMI), factoring for age and gender [1]. Regardless of age, obesity can impact various systems, including the heart, bones, and hormonal functions. Its effects are not solely physical; they also extend to mental and emotional well-being. The protein group of insulin-like growth factors plays a key role in metabolic processes [2]. Children experiencing overweight or obesity tend to have higher serum levels of insulin-like growth factor 1 (IGF-1) and insulin-like growth factor 2 (IGF-2), specifically during the phase before or at the onset of puberty [3,4]. In adulthood, IGF-1 can potentially work as a mitogen factor, and IGF-2 may contribute to greater tumor aggression [5,6].

The main aim of this study is to demonstrate the relationship between IGFs and obesity as well as its consequences.

### Family Physiology of IGFs

In 1976, two molecules were isolated from human plasma; due to their structural similarity to insulin, they were named insulin-like growth factor 1 (70 amino acids) and insulin-like growth factor 2 (67 amino acids). Several resemblances to proinsulin were noted in the structure of the two peptides: three disulfide bonds and six half-cystine residues, as well as hydrophobic amino acid residues that formed the cores of the monomers. These allowed individual peptides to maintain their three-dimensional form. Their hydrophilic surface residues differed in spacious arrangement, which may explain why the peptides reacted immunologically differently from proinsulin. Another important difference between IGF and proinsulin is the lack of double basic amino acids at the ends of the linker peptide. This makes the C-peptide produced in the formation of insulin relatively easy to enzymatically cleave from proinsulin but not from IGF proteins [7]. However, a significant effect of IGF-1 on somatic growth was discovered as early as the 1950s. Since then, IGF-1 has remained a better-researched protein than IGF-2 [8].

IGFs are secreted by most tissues; however, the majority of IGFs present in the circulation come from the liver, where their production is highly dependent on nutritional status. IGF-1 levels are variable over the course of life, rising dynamically until sexual maturity and then slowly declining throughout adulthood. Prenatally, IGF-1 levels in fetal serum gradually increase with gestational age. In postnatal life, IGF-1 operates on an axis with growth hormone [9].

The IGF-2 production rate is at its highest during prenatal life. After birth, IGF-2 levels are constant, regardless of the developmental stage or growth hormone axis [10].

The main functions of IGF-1 are to promote longitudinal growth, stimulate proliferation and protein synthesis in the majority of body cells (especially in bones and cartilage tissue), develop the nervous system (protect neurons from apoptosis), promote the proliferation and survival of pancreatic islets, and maintain stem cells. It also has an effect on the kidneys, lungs, vascular endothelium, and eyes. IGF-2 plays a major role in promoting fetal and placental growth and development. In postnatal life, it promotes skeletal muscle development and neurogenesis in the subventricular and subgranular zones of the brain, as well as in the hippocampus (which mediates memory consolidation). Its metabolic role is to differentiate adipocytes and influence glucose uptake. It maintains the beta cells of the pancreas, so it affects insulin secretion [10].

In human serum, insulin circulates in a free, unbound state, but IGFs bind with high affinity to six binding proteins: insulin-like growth factor-binding proteins 1 to 6 (IGFBP-1 to IGFBP-6). IGFBPs are closely structurally related [8]. They bind to both IGF-1 and IGF-2 with higher allegiance than IGF receptors on the cell surface, thus limiting the availability of IGF for the activation of the receptor. Unbound IGFs are eliminated rapidly, with a half-life of approximately 10 min; however, when bonded to IGFBP, their half-life is extended to approximately 30–90 min [10]. IGFBP-1 is the most predominant IGFBP in amniotic fluid, normally expressed in the liver, endometrium, and placenta. The biological action of IGFBP-1 is to inhibit tumorigenesis, mainly by binding to IGF to prevent its binding to the IGF receptor and then counteracting IGF-induced tumor growth [11]. IGFBP-2’s main function begins with binding to IGFs and modulating their action at the systemic level, stimulating cell migration and proliferation, adhesion, and cell differentiation. IGFBP-2 has a more pronounced affinity for IGF-2 than IGF-1 [12]. IGFBP-3 is the main binding partner of circulating IGFs and controls their bioavailability, and it can also exert pro-survival or proliferative, as well as pro-apoptotic, effects on neoplastic cells. Elevated levels of circulating IGFBP-3 are associated with both increased BMI and an increased risk of premenopausal breast cancer [13]. GFBP-4 is multifunctional, demonstrating IGF-independent activity in addition to its IGF-binding role. It is also degraded by a specific protease referred to as pregnancy-associated plasma protein-A (PAPP-A). Evidence suggests that IGFBP-4 and PAPP-A are involved in cancer development [14]. IGFBP-5 has the largest range of biological actions among IGFBPs. It can exert a number of biological effects, including prolonging the half-life of IGFs in the serum by concentrating IGFs in certain cells and tissues. IGFBP-5 also demonstrates IGF-independent activity [15], and IGFBP-6 is an important factor in the immune response. The IGF-independent effects of IGFBP-6 include the initiation of chemotaxis, the potential to increase oxidative burst and neutrophil degranulation, and the regulation of the Sonic Hedgehog (SHH) signaling pathway during fibrosis [16]. IGFBPs are often unbalanced in underlying pathological conditions [17].

## 2. Methods

An electronic search for literature, which was updated on 7 January 2024, was performed in PubMed. We used a combination of the following keywords: “child”, “pediatric”, “obesity”, “IGF-1”, “IGF-2”, “IGF”, “feral”, “baby”, “diabetes”, “metabolic syndrome”, “liver disease”, “cirrhosis” “liver damage”, “fatty liver”, “cancer”, “cancer promoting agent”, “ chronic lymphocytic leukemia”, “carcinogenesis”, “tumor”, “cardiovascular disorder”, “heart failure”, “cardiac hypertrophy”, and “cardiomyocyte” (Table 1). The exact search formulas can be found under PRISMA flow diagrams (Figure 1, Figure 2, Figure 3, Figure 4 and Figure 5).

We excluded all articles that were not available in English and where the full text was not accessible. A first screening was performed by automatic tools, limiting articles to the last 10 years, and a second screening was performed by reading titles and abstracts of the studies. Duplicate articles were removed. A total of 1131 items were reviewed by using the PRISMA method [18], and 252 articles were removed because of lack of free full text. The process is shown in 5 PRISMA flow diagrams (Figure 1, Figure 2, Figure 3, Figure 4 and Figure 5).

**Figure 1 ijms-25-03966-f001:**
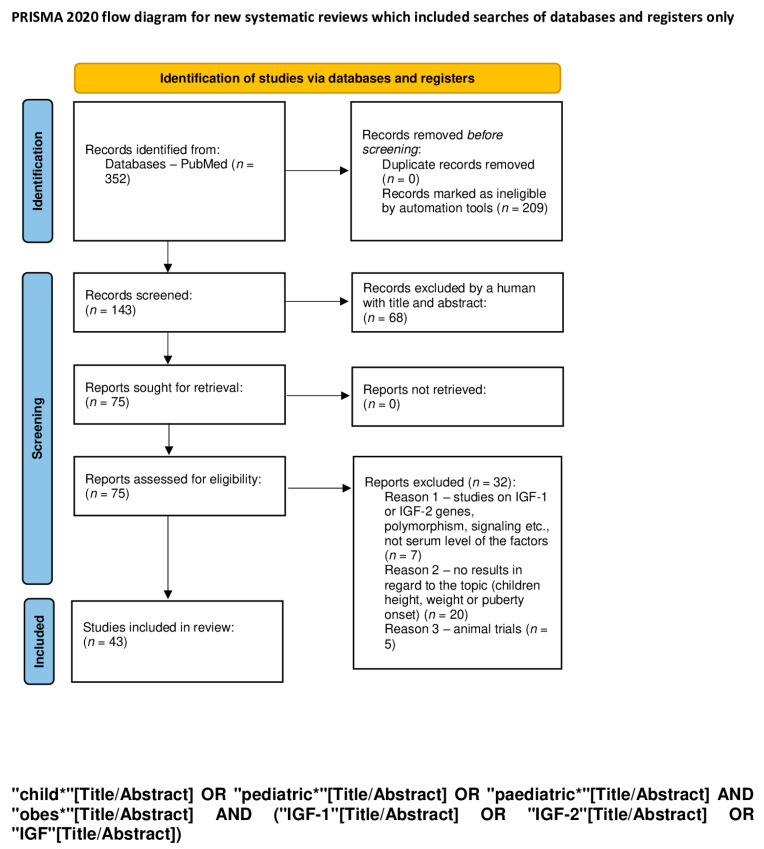
PRISMA flow diagram outlining the impacts of IGF-1 and IGF-2 on child growth (figure self-created using PRISMA pattern [18]) The asterisk (*) functions as a wildcard character that represents any string of characters. For more information, visit: http://www.prisma-statement.org/.

**Figure 2 ijms-25-03966-f002:**
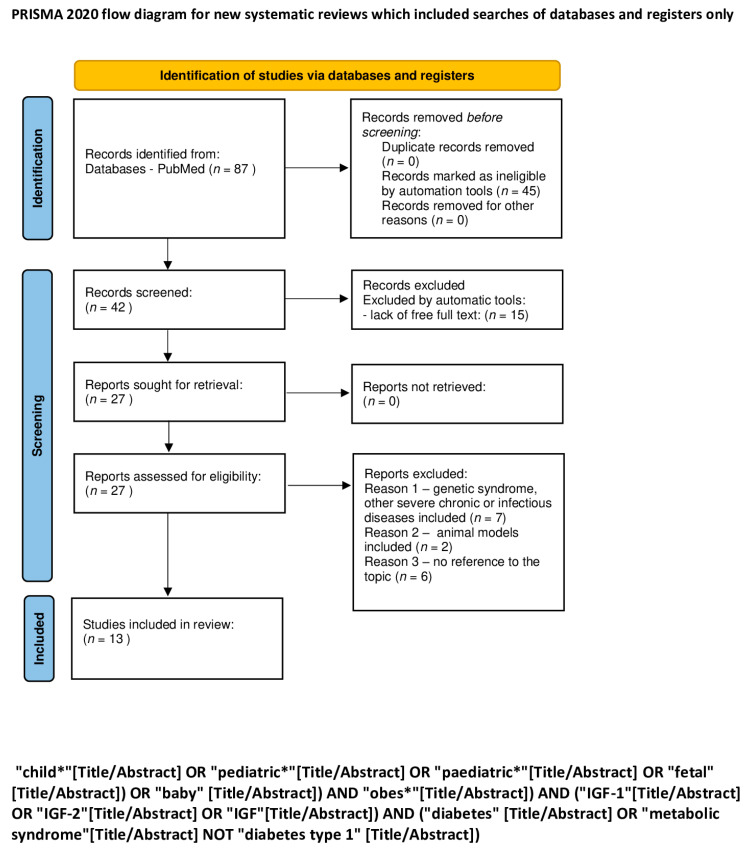
PRISMA flow diagram of IGF-1 and IGF-2 in metabolic syndrome and diabetes (figure self-created using PRISMA pattern [18]). The asterisk (*) functions as a wildcard character that represents any string of characters. For more information, visit: http://www.prisma-statement.org/.

**Figure 3 ijms-25-03966-f003:**
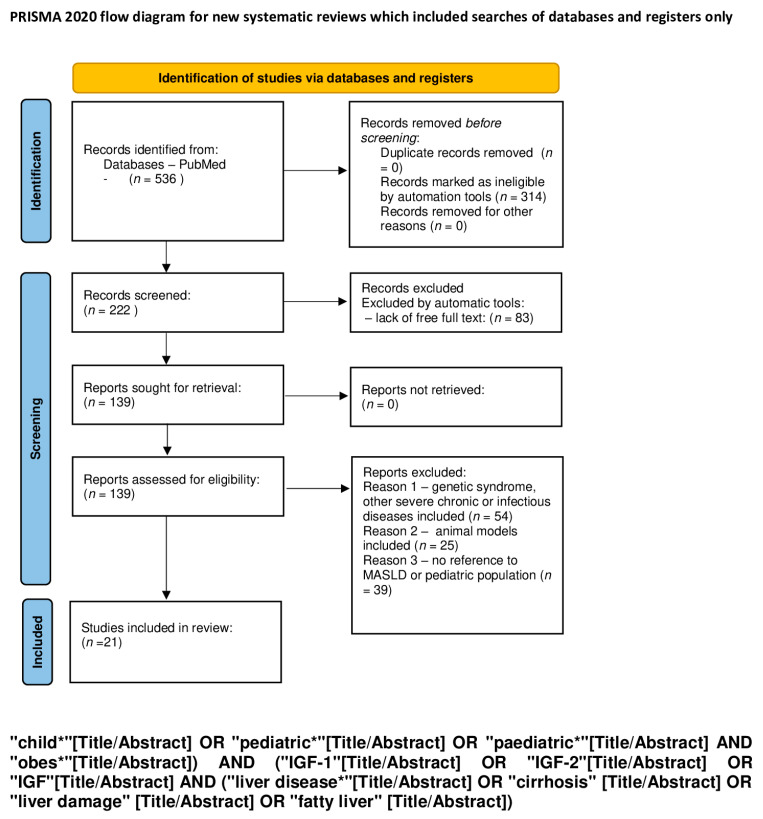
PRISMA flow diagram showing the influence of IGF-1 and IGF-2 on hepatic metabolism (figure self-created using PRISMA pattern [18]). The asterisk (*) functions as a wildcard character that represents any string of characters. For more information, visit: http://www.prisma-statement.org/.

**Figure 4 ijms-25-03966-f004:**
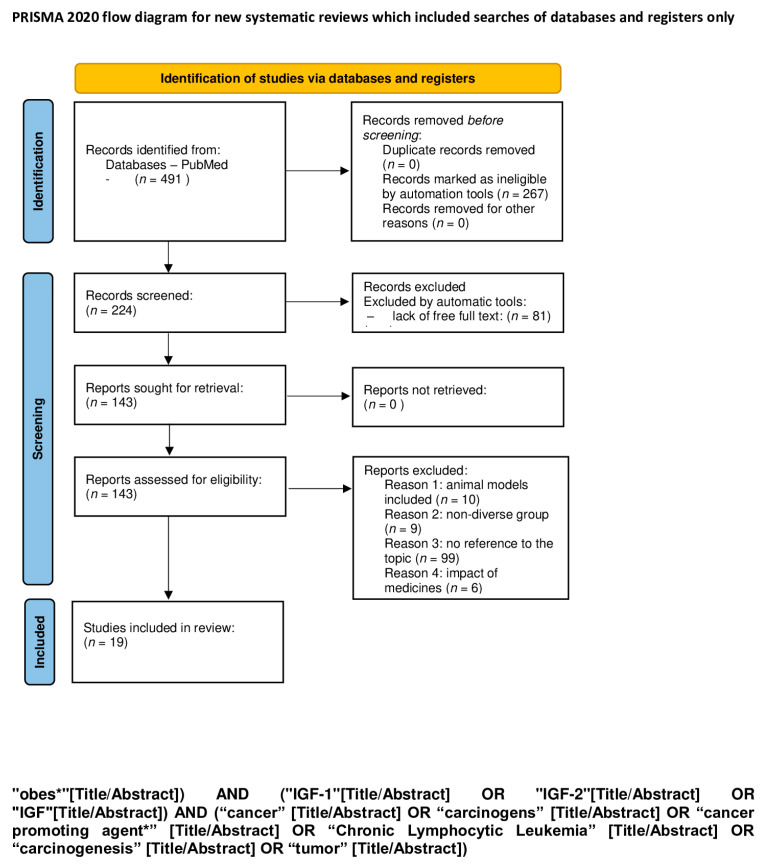
PRISMA flow diagram of IGF-1 and IGF-2 as carcinogens and cancer-promoting factors (figure self-created using PRISMA pattern [18]). The asterisk (*) functions as a wildcard character that represents any string of characters. For more information, visit: http://www.prisma-statement.org/.

**Figure 5 ijms-25-03966-f005:**
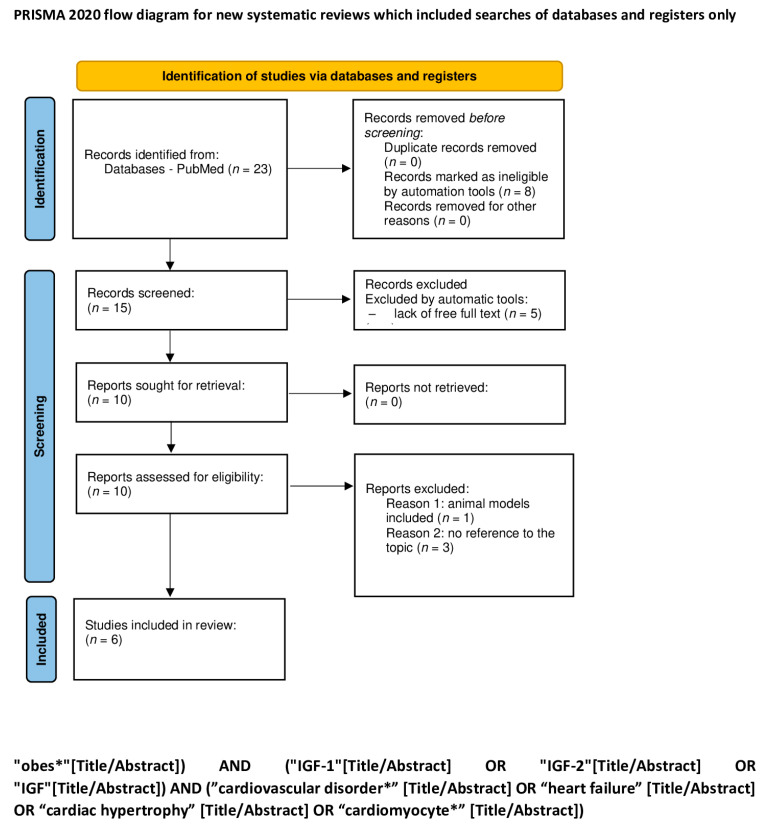
PRISMA flow diagram showing the roles of IGF-1 and IGF-2 in cardiovascular disorders (figure self-created using PRISMA pattern [18]). The asterisk (*) functions as a wildcard character that represents any string of characters. For more information, visit: http://www.prisma-statement.org/.

## 3. Results and Discussion

### 3.1. Impact of IGF-1 and IGF-2 on Child Growth 

#### 3.1.1. Fetal Life

Both IGF-1 and IGF-2 play a role in fetal growth [4,18], though the importance of the latter in the postnatal period is hard to establish due to the absence of its endocrine effects in humans [4] (Figure 6).

During pregnancy, the serum levels of IGF-1 in the fetus increase with gestational age. The higher the factor, the more intense the growth, especially in the second stage of intrauterine life [9]. The activity of placental insulin/IGF-1 axis signaling is positively correlated with the insulin level, glucose uptake, and growth and deposition of fat tissue of the fetus [22].

IGF-2 is first produced by the placenta and then by the liver. It can bind to insulin receptors and is thought to mediate insulin’s effect on adipose tissue, resulting in adipocyte growth in utero, higher birth weight, and increased fat accumulation. Higher levels of IGF-2 may also affect the size of the placenta, thus increasing nutrient delivery and fetal growth. However, data on the association between cord IGF-2 levels and children’s birth weight are inconsistent, presumably due to the fact that IGF-2 levels at birth do not always correlate with its levels during fetal development, as its levels vary throughout gestation [23].

Maternal pre-pregnancy obesity influences the metabolic profile of the child [24]. The amount of fat mass in mothers is linked to the fat mass of their newborns [25] and rapid weight gain in early childhood [26].

The data regarding correlations between maternal obesity and IGF-1 and IGF-2 in children are scarce and sometimes inconsistent. Placental insulin/IGF-1 axis signaling may contribute to the correlation between obese mothers and the occurrence of obesity and other metabolic diseases in their children [22]. Babies born to obese mothers are heavier and longer than normal-weight mothers’ offspring, though there are no differences in BMI, and the IGF-1 level in 9-month-old children is negatively associated with maternal obesity [24]. On the other hand, there is a statistically significant positive correlation between paternal adiposity and IGF-1 in offspring (measured at the age of 2 years). When stratified by sex, the percentage of fat mass in fathers and the IGF-1 blood levels in their sons are positively correlated, and there is a possibly significant correlation between maternal adiposity and IGF-1 levels in their daughters [25].

There is a correlation between a higher IGF-1 level in the cord and the weight of the newborn, with no such association for IGF-2 [4,27]. Cord IGF-1 is positively associated with infant measures of growth and body composition, such as BMI and mid-upper arm and abdominal circumference at 6 months of life. This supports the influence of in utero exposure to IGF-1—elevated, e.g., due to maternal obesity—on early growth [27]. On the contrary, in another study [26], cord blood IGF-2 was negatively associated with rapid weight gain between the first and fifth year of life, while no correlations were noted for IGF-1.

#### 3.1.2. Children’s Weight

The IGF-1 axis regulates growth, adipose tissue differentiation, and early adipogenesis in children [28]. The growth hormone (GH) control of IGF-1 is not fully developed after birth; levels of serum IGF-1 increase until approximately midpubertal age, when the growth hormone (GH)–insulin-like growth factor (IGF)-1 axis is fully developed, and eventually decline with age [9]. There is a U-shaped relationship between birth weight and future risk of obesity, with both lower and higher birth weights within the normal limits imposing a risk [9].

Data on the relationship between IGF-1 levels and weight in children are inconsistent. There are studies describing comparable [2], lower [29,30,31,32,33], or higher [3,20,34,35,36,37,38,39,40,41] IGF-1 levels and higher IGF-2 levels [4] in obese children compared to their normal-weight peers. However, IGF-1 shows a tendency to drop in children with extremely high BMI values [39]. Girls have higher IGF-1 levels than boys [42,43], probably due to a higher percentage of fat tissue. There are inconsistent reports regarding the relationship between serum IGF-1 and specific features. A positive correlation between serum IGF-1 and weight [42], BMI, waist circumference, and fat mass percentage [37] has been described. On the other hand, no association between IGF-1 and BMI [44,45], fat mass percentage [42], excess body weight, waist circumference [46], or sagittal abdominal diameter [47] has been reported. IGF-2 is significantly associated with insulin sensitivity-related parameters [4]. There is also a report of there being no association between IGF-1 in healthy 3-year-old children and early development of obesity; still, this relationship may occur later in life, as an increased tempo of growth can promote an early adiposity rebound [42].

IGF-1 levels are affected by nutritional status. Increased weight gain in infancy leads to increases in insulin and IGF-1 concentrations, which stimulate linear growth [48].

#### 3.1.3. Bone Development

The GH/IGF-1 axis is essential in the regulation of bone maturation (Figure 7). Serum IGF-1 concentration is a risk factor for significantly excessive bone age regarding chronological age [49]. Accelerated bone age (higher than chronological-age-appropriate) [19,31] and higher bone size and mass are often present in obese children [21]. IGF-1, body fat mass, osteocalcin, and BMI are predictors of increases in height and bone age in children [50], although none of these factors alone are responsible for these issues, so other variables must also be contributory [19].

Increased IGF-1 in prepubertal obese girls stimulates the maturation and proliferation of the chondrocytes as well as the mineral accretion and maturation of the bones [50]. Prepubertal children with high dehydroepiandrosterone sulfate (DHEAS) levels present more advanced bone maturation, are significantly taller and more overweight than their contemporaries, and have higher concentrations of IGF-1 [48]. There is also a positive correlation between IGF-1 and the percentage of fat mass in obese prepubertal children and the activity of bone-specific alkaline phosphatase (BALP), reflecting growth and bone mass. This supports the concept of promoting the influence of IGF-1 on accelerated bone formation [40]. No association between IGF-1 and fibroblast growth factor 21 (FGF-21) playing a role in bone metabolism has been found [51].

#### 3.1.4. Children’s Height and Its Relationship with Puberty

Children with obesity are usually taller than their normal-weight peers, which may be caused by increased circulating levels of IGF-1 [3,20,52,53], as GH release is suppressed, which implies that their height is rather GF-independent [20] or might be caused by increased adrenal androgens [54]. One study [30] reports the higher stature of obese children but also lower IGF-1 compared to a normal-weight control group. Furthermore, a higher BMI in childhood is positively correlated with a higher IGF-1 level due to associated hormones, such as insulin, that are well known as being influential on human growth [20,50] rather than being caused by excessive fat tissue [3].

The most pronounced difference in height appears around the age of 6 years in boys and 8 years in girls [53]. Obese children have higher mean IGF-1 levels compared with normal-weight subjects up to 13 years old [34] or stage 4 of thelarche [3], when the levels in both groups equalize. This suggests a height-promoting effect of the factor in obese children until puberty [3,34] and a younger age of peak height velocity [50].

During puberty, the IGF-1 in obese children decreases below the levels seen in normal-weight children, and a decreased growth velocity has been observed in obese children [53]. Likewise, sex steroids—the testosterone level for obese boys and estrogen for obese girls—also decrease during puberty, which thus might contribute to a reduced serum IGF-1 level and a lack of growth spurts [53]. Though obese children are usually taller than their normal-weight peers, this growth predominance does not persist into adult life. The potential genetic adult height of children with obesity may be lower than that of normal-weight individuals [20]; however, due to the prepubertal growth advantage, final heights are comparable among obese and non-obese adults [39,50] (Figure 6). However, impaired adult height cannot be ruled out for obese children who are short or relatively short before puberty and for children with extremely high BMI values who initially have lower IGF-1 levels [39].

### 3.2. Impact of IGF-1 and IGF-2 on Puberty in Children

Childhood obesity influences growth and early-onset puberty [20,50,55] (Figure 6). Higher insulin and IGF-1 levels in mid-childhood are associated with earlier puberty onset [20,50]—thelarche and menarche in girls and gonadarche in boys [34,52,56]—though there are reports describing no association [57]. The level of serum IGF-1 rises until about pubertal age, reaching a peak that is up to nine times higher than in 3-month-old babies [34] at about 11–16 years for girls, depending on the origin [29], with the mean level being slightly higher for boys [34] and eventually starting to decline [29,34]. This supports the statement of a correlation between the IGF-1 level and the onset of puberty, which is the main inhibitor of IGF-1 quantity. It is also possible that earlier puberty in obese girls might be a consequence of higher levels of estrogen due to increased aromatization of the androgen precursors in adipose tissue [37]. The highest levels of IGF-1 were observed in the fourth stage of the Tanner scale (TS) in both sexes [19,29,34], with peak levels for the factor being lower [34] or higher [19] in girls than in boys. Moreover, oral estrogen–progestin contraceptive drugs (CDs) result in even lower IGF-1 levels in late-pubertal girls [34]. However, there are also reports of there being no statistically significant correlation between obesity and IGF-1 levels regarding the TS, so BMI [52] and the TS should be considered when interpreting the clinical relevance of IGF-1 [34].

Appendix A contains IGF-1 and IGF-2 values (ng/mL) during child development.

### 3.3. IGF-1 and IGF-2 in Metabolic Syndrome and Diabetes 

In recent years, the number of diagnoses of type 2 diabetes in children has tremendously increased in relation to the huge rise in obesity among the youngest age group (up to 20 years of age). It is assumed that the obese pediatric population will develop or worsen obesity and its complications in adulthood, but studies suggest that some children will develop obesity complications as early as childhood. Insulin and IGF-1 signaling pathways play a role in maintaining the proper quantity and quality of pancreatic beta cells and the onset of metabolic syndrome [33,55].

Abnormal IGF/IGFBP levels have been linked to the development of glucose intolerance and metabolic syndrome. Imbalanced IGF-1 levels can induce hyperinsulinemia, while IGF-2 inhibits hepatic glucose synthesis and prevents glycogen production [9]. Elevated IGF-1 levels co-occurring with atherosclerosis lead to retinal vascular damage in obese children as early as adolescence [58]. There is a breadth of scientific evidence linking obesity to insulin resistance, hyperglycemia, and lipid disorders [58,59] (Figure 7).

Unfortunately, data from adolescents may be affected by IGF regulation by growth hormones. When insulin resistance occurs but glucose levels remain normal, there is an initial increase in free IGF-1. However, when abnormal levels of fasting glucose appear, IGF-I bioactivity reaches a plateau. Then, when glucose levels rise enough to diagnose diabetes, the bioactivity of circulating IGF-1 gradually decreases. Obesity is associated with resistance to IGF-1 at the whole-body level, and the IGF family can represent potential biomarkers of type 2 diabetes classification. Epigenetic regulation, such as DNA methylation of the IGF-2 gene, may be an important factor in determining childhood obesity. IGF-2 can also influence the tendency to have excess body weight in childhood through epigenetic regulation such as the DNA methylation of the IGF-2 gene [3,60].

It is considered that obesity may have its metabolic origins as early as fetal life. Excess body weight before or during pregnancy can affect the secretion and action of IGF-1 and IGF-2, leading to gestational diabetes. Both IGFs are elevated in both maternal and umbilical cord blood, resulting in increased fetal weight, i.e., macrosomia [61]. Maternal obesity increases the prevalence of excess weight in newborns, thereby increasing the risk of diabetes during postnatal life. There was a statistically significant correlation between the homeostasis model assessment of the insulin resistance index (HOMA-IR) and the level of IGF-2 in the cord blood of newborns born to mothers with gestational diabetes, suggesting a potential influence of IGF-2 on the development of intrauterine insulin resistance [62]. On the other hand, other studies have shown that it is low birth weight and rapid weight gain after birth that increase the risk of obesity, insulin resistance, and cardiovascular disease, and the concentration of IGFs in fetal life has a stronger effect on body length than on body weight. Lower IGF-1 levels at birth were associated with greater increases in both length and weight by the time infants reached 2 months of age. Children who experience rapid early weight gain develop an elevated BMI and insulin resistance, even as early as 8 years of age [63].

Further research into the role of IGFs during the perinatal period could be useful in predicting and preventing metabolic disorders and other serious consequences of obesity, with great health benefits in adulthood [22].

**Figure 7 ijms-25-03966-f007:**
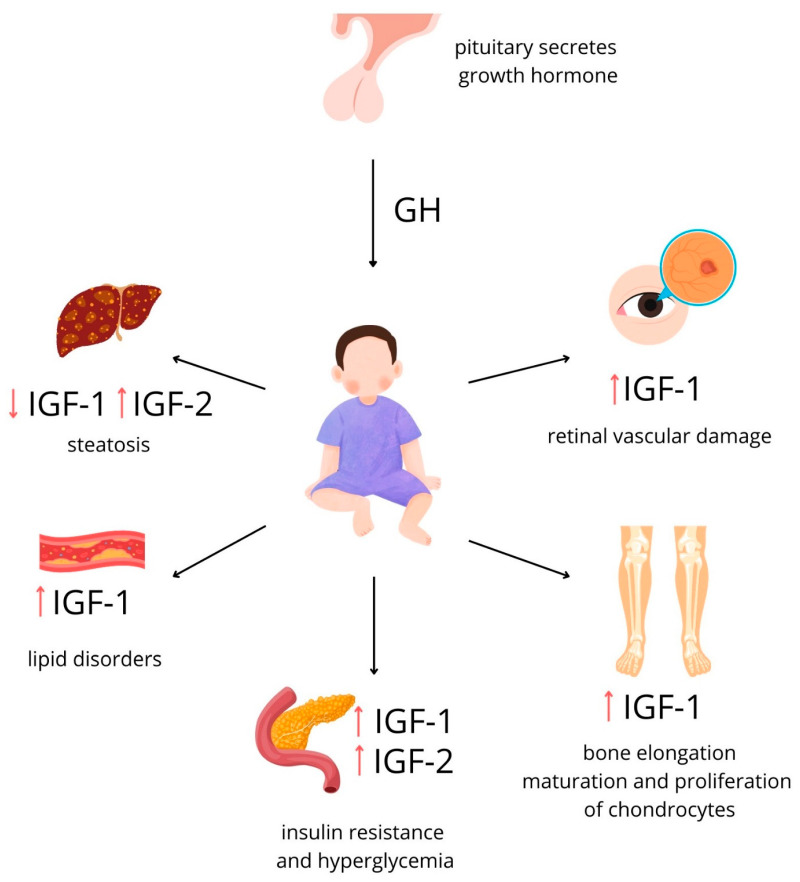
Consequences of obesity in relation to activity of IGFs. IGF-1 operates on an axis with growth hormone; IGF-2 levels are constant, regardless of the growth hormone axis [9,10]. Levels of both IGFs are elevated in obesity [3,4]. There is substantial scientific evidence linking obesity to insulin resistance, hyperglycemia, and lipid disorders [58,59]. Elevated IGF-1 levels co-occurring with atherosclerosis lead to retinal vascular damage [4]. The GH/IGF-1 axis is essential for regulating bone maturation [27]. Liver steatosis is one of the complications of obesity; it is unclear whether reduced IGF-1 levels are the cause of liver damage or, conversely, its effect [64]. (Figure self-created using Canva software; www.canva.com, accessed on 20 February 2024).

### 3.4. The Influence of IGF-1 and IGF-2 on Hepatic Metabolism

Metabolic dysfunction-associated steatotic liver disease (MASLD) is one of the most common chronic liver diseases, estimated to occur with a prevalence of 10% in children and 25% in adults worldwide, and is predicted to be the most common cause of liver transplantation by 2025 [65,66]. Most patients present no symptoms, and blood test results may remain normal [67]. Steatosis involves fatty deposits in the hepatocytes, while metabolic dysfunction-associated steatohepatitis (MASH) additionally appears with inflammatory cell infiltration, ballooning of hepatocytes, and fibrosis. The next stages of pathological liver remodeling are cirrhosis and the onset of hepatocellular carcinoma [68].

The fact is that the growing number of patients with MASLD is related to the obesity epidemic; the insulin resistance that occurs in its course is a significant risk factor for MASLD, leading to the development of oxidative stress and lipotoxicity [65]. A large body of evidence has demonstrated the impact of IGF-1/GH axis malfunctions as a possible contributor to the development of MASLD and MASH [69,70,71,72]. GH is known to stimulate IGF-1 levels and insulin concentrations, primarily by initiating insulin resistance. Insulin levels in the portal vein can control the production of IGF-1 by GH [73].

The primary functions of the IGF system in liver physiology include its roles in organ development, growth, and regeneration [65]. Studies in animals have shown anti-inflammatory and anti-fibrotic effects of IGF-1 and GH on the liver [70,74].

Deletion of the IGF-1 gene in the liver results in insulin resistance, suggesting that hepatic IGF-I regulates systemic insulin sensitivity [68]. Because of its potential lipolytic, anti-inflammatory, and immunomodulatory effects, the GH/IGF-1 axis is a potential disease-modifying goal in MASLD [64] (Figure 7).

However, different researchers have come to opposing conclusions. Some studies have confirmed a significant relationship between IGF-1 and GH levels and the severity of MASLD [16,23,75]. Research has shown that decreased IGF-1 levels are related to radiological and histological exponents of MASLD, consisting of lobular inflammation, increased hepatocyte volume, MASH, and fibrosis [70,76,77]. An association between low serum IGF-1 levels and increased histological severity of MASLD has also been indicated [75,78]. Others found no correlation between serum IGF-1 levels and the severity of hepatic steatosis and fibrosis [74].

It is unclear whether reduced IGF-1 levels are the cause of liver damage or, conversely, its effect [64]. In clinical practice, screening for MASLD is recommended for people with metabolic risk factors, such as obesity, type 2 diabetes, and hypertension [67]. It is possible that reduced IGF-1 corresponds to a decreased capacity for synthesis in the liver and leads to MASLD-related sarcopenia in the course of sarcopenic obesity. It involves the loss of muscle mass and the accumulation of ectopic fat in the liver [79]. GH and IGF-1 supplementation induce significant improvement in both hepatic steatosis and sarcopenia, allowing the conclusion to be drawn that IGF-1, as one of many factors, has a pleiotropic effect in liver disease. In conclusion, IGF-1 has the ability to regenerate a damaged liver [71].

IGF-2 also has an impact on the course of MASLD. Studies have shown that plasma levels of this factor in obese children are reflected in the histological progression of the disease. An increase in IGF-2 levels has been observed, which is consistent with the development of both steatosis and fibrosis [80].

Hepatocellular carcinoma (HCC) is a type of cancer that develops as a consequence of hepatic steatosis and fibrosis (Figure 8). Obesity, cirrhosis, pre-diabetes, and type 2 diabetes, among other conditions, predispose individuals to the development and progression of HCC. About 35% of cases of non-alcoholic fatty liver disease lead to liver fibrosis and potentially HCC. Up to 20% of patients with MASLD and type 2 diabetes will develop liver fibrosis during their lifetime [66]. IGFs might be potential biomarkers and therapeutic targets for MASLD and HCC [80,81,82]; IGF-1 can be used to identify advanced fibrosis, particularly when correlated with the international normalized ratio (INR) and ferritin, and corresponds with alterations in serum lipids associated with fibrosis [67,83]. Another promising biomarker from the IGF family is IGFBP-1. A specific and sensitive correlation between protein secreted in the liver under the influence of insulin and the development of hepatic insulin resistance has been shown [84].

### 3.5. IGF-1 and IGF-2 as Carcinogens and Cancer-Promoting Factors 

Nearly one in five children in the United States are considered obese. It is predicted that more than half of these children will be obese as adults. Many factors contribute to early childhood obesity, including maternal obesity. This leads to the phenomenon of intergenerational obesity and its many consequences for children and the adults they become [85]. IGF-1 is implicated in the development and progression of various types of cancer. It is a potent mitogen that has an anti-apoptotic function and promotes the formation of cancer metastases. Its high concentration in serum may accelerate cell proliferation, which could lead to the growth and evolution of colorectal, breast, and prostate cancer [5] (Figure 8). This occurs through mediating between the tumor and stroma in the microenvironment through paracrine signaling [86]. Elevated IGF-1 levels may be linked to a higher risk of ovarian cancer in women of reproductive age [87] (Figure 8). IGF-1 binds to its receptor—insulin-like growth factor 1 receptor (IGF-1R)—which activates oncogenic pathways. Elevated levels of circulating insulin-like growth factor-1 (IGF-1) have been consistently associated with an increased risk of prostate cancer, complications, and mortality [88]. IGFs are hormones associated with the growth, survival, apoptosis, and migration of multiple myeloma cells. IGF-1 is crucial for the survival of malignant cells of hematopoietic origin, as it inhibits apoptosis and induces cellular transformation. Furthermore, IGF-1R promotes the survival of unanchored cancer cells, facilitating tumor dissemination. The level of IGF-1 may also serve as a predictor of disease progression. It is important to note that cancer cells do not secrete IGF-1. Excessive synthesis of IGF-1 by the liver is associated with an increased risk of cancer in both children and adults and contributes to the maintenance and progression of cancer. A higher concentration of IGF-2 may be correlated with greater tumor aggression (Figure 8). There may be a link between increased IGF-1 levels in overweight and obesity and the development of CLL (chronic lymphocytic leukemia) [6,89] (Figure 8). Excess body weight and insulin resistance are also contributing factors in the development of esophageal adenocarcinoma. Patients affected by this condition have been observed to have high levels of IGF-1. The activation of the IGF receptor also contributes to esophageal carcinogenesis, even in the absence of hyperinsulinemia, when duodenal reflux is present [90,91] (Figure 8). most solid malignancies secrete IGF-2. Signals can be transmitted through various means, including autocrine signaling, which results in a self-stimulating effect in cancer. This process enables the evasion of apoptosis and promotes growth and proliferation. IGF-2 has a wider range of opportunities to regulate and control transcription at the gene promoter level compared to IGF-1. In some cases, cancers use insulin-like growth factor 2 as an autocrine growth factor, which exacerbates the malignant characteristics of the tumor. Tumors secrete IGF-2, which weakly interacts with soluble extracellular proteins of the IGFBP family. IGF-2 is responsible for hypoglycemia in the paraneoplastic syndrome of patients with solid malignancies. Research has demonstrated that the overexpression of IGF-2 is linked to cancers that are only marginally neutralized by the immune system, such as malignant mesothelioma and glioblastoma multiforme [89,92]. Epidemiological studies indicate that an increase in BMI is directly associated with an increase in the risk of pancreatic cancer due to elevated serum IGF-1 levels. In addition, a more aggressive disease course and shorter overall survival are associated with high IGF-1R expression. The study participants’ mean biomarker levels were as follows: IGF-I levels were 201.47 ng/mL for men and 155.43 ng/mL for women, and IGF-II levels were 1585.46 ng/mL for men and 1715.08 ng/mL for women [93,94]. Obesity is also associated with thyroid and endometrial cancers. Studies have shown that elevated levels of insulin and IGF-1 are associated with multistep endometriosis and endometrial carcinogenesis [95,96]. IGF-1 can promote mitosis, leading to abnormal hyperplasia, differentiation, and apoptosis of thyrocytes. This pathway is a thyroid-stimulating hormone (TSH)-independent mechanism of thyroid nodule formation in metabolic syndrome [97]. The interaction between insulin and the IGF system may also be crucial in renal cell carcinoma progression, as IGF-1 stimulates tumor angiogenesis due to increased vascular endothelial growth factor (VEGF) levels [98,99,100]. Research indicates that elevated serum IGF-1 levels are associated with an increased risk of multiple cancers. However, it has been observed that patients with gastric cancer have lower serum IGF-1 levels than healthy patients, which contradicts previously reported information about the IGF pathway [101].

**Figure 8 ijms-25-03966-f008:**
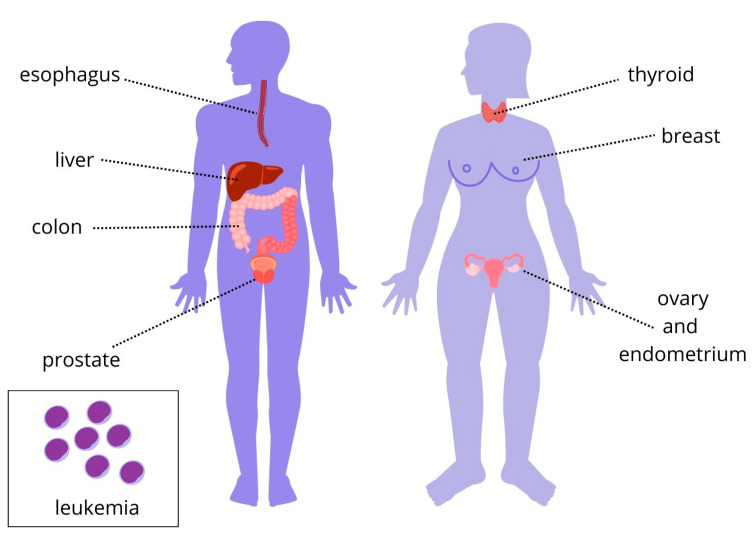
Cancers whose development is related to obesity and IGF levels. IGF-1 is implicated in the development and progression of various types of cancer. It is a potent mitogen that has an anti-apoptotic function and promotes the formation of cancer metastases [5]. Cancers that have been linked to elevated serum IGF levels include colorectal, esophageal, thyroid, and hepatic cancer and chronic lymphocytic leukemia in both sexes; breast, ovarian and endometrial cancer in women; and prostate cancer in men [5,6,23,66,87,89,90,91,95,96]. A higher concentration of IGF-2 may be correlated with greater tumor aggression [6,89] (figure self-created using Canva software; www.canva.com, accessed on 20 February 2024).

### 3.6. The Role of IGF-1 and IGF-2 in Cardiovascular Disorders 

Non-standard levels of IGF-1 and IGF-2 have an impact on the cardiovascular system [102]. IGF-1 is a molecule that triggers signaling pathways, leading to a degree of physiological hypertrophy of cardiomyocytes. Cardiac hypertrophy is an autonomous risk factor for heart failure. The signaling pathways connected to hypertrophy are closely intertwined, and physiological hypertrophy may occur in a pathological state and vice versa, depending on specific conditions. In a group of athletes, the concentrations of IGF-1 were increased, which was potentially connected to physiological hypertrophy of the heart, which raises its efficiency [103]. IGF-1 impacts blood vessel function, producing effects that can be both positive and negative. Studies have demonstrated its anti-inflammatory and anti-apoptotic properties and the potential for the stimulation of angiogenesis. Moreover, its properties encompass the generation of nitric oxide in the endothelium, which may subsequently facilitate enhanced myocardial contractility during exercise. According to certain experts, there is evidence to suggest that IGF-1 has an effect on the cardiovascular system by boosting the responsiveness of cardiac cells to insulin [104]. The effect of IGF-1 on cardiomyocytes is well demonstrated in acromegaly, a pathological condition of the overproduction of GH and IGF-1. Patients suffering with this condition may experience several adverse cardiac effects, first morphological and then functional. Hypertrophy, due to the relative elongation of cardiac myocytes, allows the sarcomeres to overlap [105,106]. IGF-2 is an anabolic factor that promotes cell proliferation and plays a role in fibrotic processes. Neonatal blood circulation of IGF-2 can encourage the proliferation and hypertrophy of cardiomyocytes. Excessive activation of the mTOR (mammalian target of rapamycin) signaling pathway is the cause of this phenomenon [107]. Studies on rats have demonstrated that increased levels of IGF-2, resulting from receptor disruption, lead to cardiomyocyte proliferation; this can also decrease hypoxia-induced apoptosis in rat cardiomyocytes [107].

### 3.7. Discussion

The complex involvement of insulin-like growth factors (IGF-1 and IGF-2) in various physiological processes spanning from fetal life to adulthood underscores their pivotal roles in human development and health. A comprehensive review of the literature highlights the multifaceted impact of these molecules on metabolic pathways, growth regulation, hormonal functions, and disease pathogenesis, particularly in the context of pediatric obesity.

Understanding the dynamic interplay of IGF-1 and IGF-2 in metabolic disorders, cancer development, cardiovascular anomalies, and other health conditions provides valuable insights into potential diagnostic and therapeutic methods. Elevated levels of IGFs in childhood obesity and their correlations with the onset of puberty, metabolic syndrome, and diabetes underscore their significance as potential biomarkers for early disease prediction and classification.

In prenatal life, an increase in IGF-1 levels is associated with increased birth weight, and its increased levels can then accelerate weight gain in infancy. The continuation of this pattern of dependence during later years of development can lead to the persistence of excessive body weight and earlier initiation of puberty. Considering the function of IGFs in development, it can be hypothesized that the child enters a vicious cycle of excessive weight and elevated IGFs. Once in this pattern, it is hard to change because excessive weight gain increases IGFs, and elevated IGFs promote the development of metabolic syndrome, which makes weight loss difficult. Such a metabolic state persisting through childhood and adolescence leads to children entering adulthood in metabolic imbalance. At this stage, the young adult has already had at least a few years of exposure to elevated levels of IGFs; when this period is prolonged, the risk of complications of obesity, particularly type 2 diabetes, the onset of cancer, and cardiovascular issues, increases.

Research on the factors involved in obesity is extremely important in the developing world because excessive body weight and its complications are becoming one of the leading causes of death in developed countries. Today, the exploration of new aspects of the onset and course of obesity is in high demand because it affects the health and lives of numerous people.

## 4. Conclusions

This study, in a novel way, tracks the actions and impacts of IGF-1 and IGF-2 on the human body over the course of a lifetime, considering disturbed metabolism in the course of obesity. Unfortunately, the current state of knowledge does not properly explain the relationship between IGFs and obesity; there are many potential directions for research in this field. In conclusion, the insulin-like growth factors IGF-1 and IGF-2 are crucial players in various physiological processes, exerting their influence from the prenatal period to adulthood.

In the context of children’s growth, both IGF-1 and IGF-2 contribute significantly, influencing fetal development, birth weight, and postnatal growth trajectories. However, the relationship between maternal obesity and IGF levels in children is complicated, with varying results and potential long-term metabolic consequences. Imbalances in IGF levels are linked to insulin resistance, glucose intolerance, and obesity-related complications such as metabolic syndrome and diabetes. IGF-1 and IGF-2 also impact hepatic metabolism, contributing to MASLD, yet conflicting findings highlight the complexity of these relationships. IGFs act as potential carcinogens, elevating malignancy risks, warranting further research into their roles in cancer biology.

In essence, the multifaceted roles of IGF-1 and IGF-2 underscore their significance in human physiology, growth, and health while also presenting challenges and opportunities for understanding and managing various health conditions across the span of life.

## 5. Future Directions

Continued research in this field holds the promise of uncovering novel diagnostic and therapeutic methods or even preventive strategies for a spectrum of disorders influenced by the IGF system. An interesting direction for further research may be the relationship between IGF-1 and IGF-2 levels and the progression of childhood obesity, as well as their impact on the frequency and severity of complications brought on by excessive body weight.

## Figures and Tables

**Figure 6 ijms-25-03966-f006:**
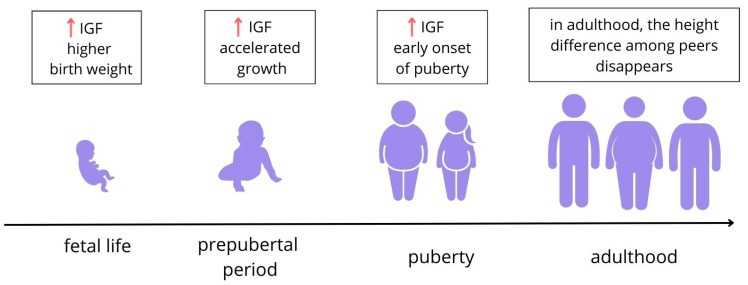
The impact of excessive body weight on growth and development throughout life. IGF-1 and IGF-2 impact the growth and maturation of tissues. In fetal life, IGFs stimulate growth; elevated levels of IGFs can lead to excessive birth weight and a higher proportion of fat in the general composition of the body [1,19]. During the prepubertal period, obese children tend to be taller than their normal-weight peers, which may be related to elevated serum IGF-1 levels [7,11,13,20]. Higher levels of insulin and IGF-1 in the first years of life are associated with a faster onset of puberty [7,21]. Growth predominance does not occur in adult life [7] (figure self-created using Canva software, www.canva.com, accessed on 20 February 2024).

**Table 1 ijms-25-03966-t001:** The keywords used for identification of studies; the exact search formulas can be found under PRISMA flow diagrams (Figure 1, Figure 2, Figure 3, Figure 4 and Figure 5). The asterisk (*) functions as a wildcard character that represents any string of characters.

Paragraph	Search Terms
Impact of IGF-1 and IGF-2 on children’s growth (Figure 1)	child* OR pediatric* OR paediatric* AND obes* AND IGF-1 OR IGF-2 OR IGF
IGF-1 and IGF-2 in metabolic syndrome and diabetes (Figure 2)	child* OR “pediatric* OR paediatric* OR fetal OR baby AND obes* AND IGF-1 OR IGF-2 OR IGF AND diabetes OR metabolic syndrome NOT diabetes type 1
The influence of IGF-1 and IGF-2 on hepatic metabolism (Figure 3)	child* OR pediatric* OR paediatric* AND obes* AND IGF-1 OR IGF-2 OR IGF AND liver disease OR cirrhosis OR liver damage OR fatty liver
IGF-1 and IGF-2 as carcinogens and cancer- promoting factors (Figure 4)	obes* AND IGF-1 OR IGF-2 OR IGF AND cancer OR carcinogens OR cancer promoting agent* OR Chronic Lymphocytic Leukemia OR carcinogenesis OR tumor
The role of IGF-1 and IGF-2 in cardiovascular disorders (Figure 5)	obes* AND IGF-1 OR IGF-2 OR IGF AND cardiovascular disorder* OR heart failure OR cardiac hypertrophy OR cardiomyocyte*

## Data Availability

Data derived from public domain resources.

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
