# Peer review of "IGF-1 and IGF-2 as Molecules Linked to Causes and Consequences of Obesity from Fetal Life to Adulthood: A Systematic Review"

_ijms, 2024, doi:10.3390/ijms25073966_

Round 1

Reviewer 1 Report

Comments and Suggestions for Authors

The authors conducted a Systematic Reviews and Meta-Analyses (PRISMA) method to show the association of child health, obesity, and insulin-like growth factors. The study reveals 14 associations between insulin-like growth factor 1 and birth weight, early growth, and adiposity.

This review article is well written and organized and can be published after a few minor revisions.

Minor concerns:

Introduction: aims are missing.

Statistics: How the author treated the bias of selected papers?

A few typos or editing errors: line 38, IGF-2);

Acronyms should be used correctly. For example, GH, IGF-1, DHEAS, MASH, MASLD, HOMA-IR, HCC, TSH, and VEGF, etc...

Reviewer 2 Report

Comments and Suggestions for Authors

In their review article, the authors have summarized in detail the current state of knowledge regarding the role of two Insulin-like growth factors in the development and course of childhood obesity.  The article is well planned, taking into account all stages of child development and also the effects of IGF-1 and IGF-2 on the functioning of individual organs. The topic due to social factors is extremely timely which makes it reasonable to publish the paper in IJMS.

A few minor comments are intended to improve the substantive aspect of the paper.

1. please explain why the authors used only one database (PubMed). After all, there are other databases SCOPUS, Web of Science. Their use would certainly improve the pool of articles with selected keywords. It should be made clear how much the exclusion of texts not available in the full version (line 122) limited the pool of articles used in the current review.

2 Whether liver, bone or eye (lines 63, 66, 330) are organs, but not tissues. In order to maintain histological accuracy, this should be changed. Furthermore, do the authors see the difference between "fat" and "adipose tissue"?

3. I don't understand the idea of Discussion in the review article (line 460). What specifically do the authors want to discuss in this extremely short chapter? After all, by definition, a review article is a form of discussion.

4. Please check the order in which abbreviations are introduced. E.g., the abbreviation MASLD first appears in line 314, but its development is only in line 492.

Reviewer 3 Report

Comments and Suggestions for Authors

Current review article conducted the associations between insulin-like growth factor 1 (IGF-1) and birth weight, early growth, and adiposity. Please check the concerns below.

1.      Main results were not indicated in the abstract. It is unclear about the imbalances in IGF levels.

2.      Obesity was indicated in the title. However, introduction section did not conduct the obesity. Why?

3.      In line 125, items were reviewed by using PRISMA method that needs reference(s) to support.

4.      In line 125, 1131 items were reviewed. But, in Figure 1, database shown 352 only. Why? Please explain this in the methods.

5.      In line 366, IGF-1 and IGF-2 as carcinogens and cancer promoting factors that shall be introduced in the introduction section.

6.      In line 453 and 454, two sentences need the reference(s) to support.

7.      The dynamic interplay of IGF-1 and IGF-2 in obesity is extremely important. Please add more in the discussion section.

8.      In conclusion, IGF-1 and IGF-2 as carcinogens and cancer promoting factors were ignored. Why?

9.      Novelty and limitation(s) in current review may strengthen it.

Comments on the Quality of English Language

It seems better to check through a professional editing service.

Reviewer 4 Report

Comments and Suggestions for Authors

This is an interesting paper, in which the authors reviewed the effect of IGF-1 and IGF-2 on obesity and other diseases. Here are some concerns that need to be addressed:

1)         The paper should concentrate on describing the findings/concepts to the readers in a clear way, so PRISMA flow diagrams can be put as supplemental figures. Is it possible to have a figure illustrating the level (with real number) and changes of the IGF-1 and IGF-2 during the different stages of life? ie., prenatal, postnatal, infant, toddler, and children until puberty. An illustration showing the level and changes will be more visual and easy for the readers to follow.

2)        Please define the abbreviations when first in use.

For example, Line 314: MASLD is one of the most common chronic liver diseases,  please define MASLD when you first use it. Line 355:  HCC please write the full name and define HCC when using it for the first time. There are more in the paper, please go through them carefully.

3)        Some sections are very hard to understand, please go over them and try to use simple sentences/words to convey the idea.

Line 232:  Though obese children are usually taller than their normal-weight peers, this growth predominance does not occur in adult-life, thus potential adult high in obese children is somehow impaired [28], but is usually not affected children due to prepubertal growth advantage [34,47].

This sentence is very hard to follow, not sure what the authors are trying to say. What is “impaired adult height”?  

Line 179: Positive correlation between serum IGF-1 and weight [37], BMI, waist circumference and percentage fat mass [32], as well as no association between the hormone quantity and BMI [39,40], fat mass percentage [37], excess body weight, waist circumference [41] or sagittal abdominal diameter [42] is reported.

This section is also very hard to follow, please clarify.

4)        In the paper, the pro-growth effect and cancer causing effect of IGFs were discussed. It would be more informative to list the range of effective levels when discussing each effect.

Same for the effect for metabolic disorders and cardiovascular disorders, at what levels of the IGFs, pathological effect was seen.

5)        This paper could benefit from some language edition, following are some specific concerns regarding use of words.

Line 128: 3.1. IGF-1 and IGF-2 – impact on children growth (fig. 1)

Impact on children’s growth

Line 314: MASLD is one of the most common chronic liver diseases,  please define MASLD when you first use it.

Line 335: opposite conclusions

Conflicting conclusions

Line 424: Cancers whose development is related to obesity and IGFs levels

The effect of obesity and IGF levels on cancer development

Comments on the Quality of English Language

1)        Some sections are very hard to understand, please go over them and try to use simple sentences/words to convey the idea.

Line 232:  Though obese children are usually taller than their normal-weight peers, this growth predominance does not occur in adult-life, thus potential adult high in obese children is somehow impaired [28], but is usually not affected children due to prepubertal growth advantage [34,47].

This sentence is very hard to follow, not sure what the authors are trying to say. What is “impaired adult height”?  

Line 179: Positive correlation between serum IGF-1 and weight [37], BMI, waist circumference and percentage fat mass [32], as well as no association between the hormone quantity and BMI [39,40], fat mass percentage [37], excess body weight, waist circumference [41] or sagittal abdominal diameter [42] is reported.

This section is also very hard to follow, please clarify.

This paper could benefit from some language edition, following are some specific concerns regarding use of words.

Line 128: 3.1. IGF-1 and IGF-2 – impact on children growth (fig. 1)

Impact on children’s growth

Line 314: MASLD is one of the most common chronic liver diseases,  please define MASLD when you first use it.

Line 335: opposite conclusions

Conflicting conclusions

Line 424: Cancers whose development is related to obesity and IGFs levels

The effect of obesity and IGF levels on cancer development

Round 2

Reviewer 3 Report

Comments and Suggestions for Authors

It has been revised mainly according to comments. However, the language checking needs gthe certificate to support.

Comments on the Quality of English Language

Language checking needs the certificate.

Author Response

The certificate of english editing is sent in attachment 
